# FontFusionGAN: Refinement of Handwritten Fonts by Font Fusion

Avinash Kumar [ID], Kyeolhee Kang, Ammar ul Hassan Muhammad and Jaeyoung Choi * [ID]

School of Computer Science and Engineering, Soongsil University, Seoul 06798, Republic of Korea; kumar@soongsil.ac.kr (A.K.); kyeolhee@soongsil.ac.kr (K.K.); ammar91@soongsil.ac.kr (A.u.H.M.)
* Correspondence: choi@ssu.ac.kr

**Abstract:** Handwritten fonts possess unique expressive qualities; however, their clarity often suffers because of inconsistent handwriting. This study introduces FontFusionGAN (FFGAN), a novel method that enhances handwritten fonts by mixing them with printed fonts. The proposed approach leverages a generative adversarial network (GAN) to synthesize fonts that combine the desirable features of both handwritten and printed font styles. Training a GAN on a comprehensive dataset of handwritten and printed fonts enables it to produce legible and visually appealing font samples. The methodology was applied to a dataset of handwriting fonts, showing substantial enhancements in the legibility of the original fonts, while retaining their unique aesthetic essence. Unlike the original GAN setting where a single noise vector is used to generate a sample image, we randomly selected two noise vectors, $z^1$ and $z^2$, from a Gaussian distribution to train the generator. Simultaneously, we input a real image into the fusion encoder for exact reconstruction. This technique ensured the learning of style mixing during training. During inference, we provided the encoder with two font images, one handwritten and the other printed font, to obtain their respective latent vectors. Subsequently, the latent vector of the handwritten font image was injected into the first five layers of the generator, whereas the latent vector of the printed font image was injected into the last two layers to obtain a refined handwritten font image. The proposed method has the potential to improve the readability of handwritten fonts, offering benefits across diverse applications, such as document composition, letter writing, and assisting individuals with reading and writing difficulties.

**Keywords:** font style fusion; font style mixing; font generation; refining handwriting; AdaIN

## 1. Introduction

Handwritten fonts are a unique and expressive communication method. They can add a personal touch to documents, letters, and other creative projects. However, they can be challenging to read, particularly if the handwriting is unclear or inconsistent. Creating these fonts is difficult and time-consuming, particularly in languages with many complicated characters. For example, the Korean language has 11,172 Hangul characters, and the Chinese language has over 50,000 Hanja characters. Recent advances in generative models have led to the development of new font synthesis methods [1–9] that use generative adversarial networks (GANs). These methods approach the font synthesis problem as an image-to-image translation task, with training in either a supervised setting with paired image data or a set level of supervision for font style labels [1–9].

Image-to-image translation is a task aimed at transferring a given source image into the target domain of reference images. Initial methods [10–14] extensively used GAN and produced visually compelling results. However, these approaches were limited in their ability to convert source images into specific, predetermined domains or categories, which constrained their practical utility. Recent advancements in this field include the introduction of few-shot methods, such as those mentioned in references [9,15–19]. The innovative techniques disentangle the content of any style of an image, enabling them

to convert a source image into a wide range of styles, as long as a few reference images are provided.

Furthermore, the RG-UNIT [20] proposed an image retrieval strategy to facilitate domain transfer. This approach involves finding images in the target domain that share content similarities with the source image and then using their content features to generate more realistic translated images. However, this retrieval strategy has limitations when applied to font generation tasks because the retrieved images may still exhibit substantial content differences from the target, particularly in fine-grained tasks such as font generation.

Although the existing methods for generating realistic fonts have achieved impressive results, they have certain limitations. First, they rely primarily on large sets of paired training samples. Second, additional fine-tuning processes [7] or techniques are often required to learn unseen font styles effectively [21,22]. Third, its applicability is restricted to specific language characteristics. Finally, most existing methods perform style transfer (i.e., transferring the style of the reference image to the content image) rather than actual style mixing (i.e., mixing desirable features of both content and style). In style transfer, we lose the unique qualities and styles of the content images.

To address these limitations, we propose a simple yet highly effective approach for enhancing handwritten font images using a style mixing technique. The methodology employed in this study draws inspiration from the mixing regularization strategy employed in StyleGAN [23], which is a cutting-edge approach for generating high-resolution human-face images. Using mixing regularization, we facilitate style localization by generating images during training using two random latent codes instead of one. This study aims to improve the readability of handwritten fonts by combining them with printed fonts, thereby addressing the clarity issues associated with inconsistent handwriting. Most current research focuses on font generation tasks, primarily on printed fonts. However, our study places a predominant emphasis on refining handwritten fonts, and, to the best of our knowledge, it is the first study to address these challenges.

In this study, we propose FontFusionGAN (FFGAN), a method for refining handwritten fonts by mixing them with printed fonts. Our approach uses a GAN to generate new fonts that combine the desirable features of both handwritten and printed fonts, as well as learn to generate new fonts that are visually appealing and legible. We evaluated our method on a dataset of handwritten fonts and found that it could generate new font images that were significantly better than the original handwritten fonts while retaining their unique style. Our model can improve the readability of handwritten fonts through font fusion. It can also be used to create new fonts for documents, letters, and creative projects and to help people with dyslexia or reading difficulties.

Our experiments demonstrate that the proposed approach can generate high-quality mixed-font images with a variety of styles. We also demonstrate that our approach can be used to refine existing handwritten fonts, making them more visually appealing. We believe that our approach is a promising method for font fusion and improving the quality of handwritten font images. It has the potential to make font creation more efficient and easier to use and can be used to create a wider variety of mixed fonts.

FFGAN enhances the StyleGAN-V2 [24] model by developing a fusion encoder network and removing the mapping network. The addition of this functionality enables FFGAN to learn the process of converting a given font image into an appropriate latent code vector that represents the image-style code. The fusion encoder successfully executes the projection process in real time, generating a latent code vector representation that can be incorporated into the StyleGAN generator. Consequently, FFGAN enables controllable style mixing to refine handwritten font images during inference. Using the inject index property, we input a handwritten font image into the initial five layers of the generator and a printed font image into the last two layers of the generator, which shows better-quality fusion results.

By contrast, image-to-image (I2I)-based font synthesis methods employ separate encoders and multitask discriminators for style and content learning [10,25]. FFGAN

jointly learns how to encode printed and handwritten images into a latent code vector and then mixes them in a real-time scenario using a trained generator. This simple yet effective strategy enables FFGAN to fuse previously unseen font styles in a one-to-one image-mapping setting during inference without any architectural constraints or additional training requirements. Additionally, because FFGAN operates in an unsupervised fashion, it eliminates the need for content or class-label-based supervision. Figure 1 shows some of the results of our model.

**Figure 1.** Results of refining the handwritten font images by merging them with printed fonts.

A series of experiments was conducted to assess the validity of our proposed methodology through both quantitative and qualitative measures. Moreover, by integrating our unsupervised methodology with a unique training technique, FFGAN exhibits the potential for seamless expansion into many text-image-related tasks. Some of these include style transfer, font attribute manipulation, and the development of random font styles. These extensions demonstrated the potential of the proposed strategy.

## 2. Methodology

### 2.1. FFGAN Architecture

The architectural framework of FFGAN is shown in Figure 2a. The framework consists of three primary elements: a fusion encoder $E$, generator $G$, and discriminator $D$.

The initialization process of generator $G$ includes the use of a constant vector with dimensions of $4 \times 4 \times 64$. This vector is used to describe the spatial properties of the initial layer. The proposed methodology enables generator $G$ to obtain its style vector from two distinct sources. First, as shown in the upper box in Figure 2a, two latent codes, $z^1$ and $z^2$, are sampled from a Gaussian distribution.

Furthermore, as shown by the lower box in Figure 2a, we pass a real image x to a fusion encoder $E$, resulting in the output of a style vector represented as $W$. Subsequently, using adaptive instance normalization (AdaIN) [18], we transform and inject the latent codes ($z^1$, $z^2$) or style code ($W$) into each convolutional layer of generator $G$.

Our methodology is different from current and previous font generation approaches, which employ a multitask discriminator to generate realistic images and ensure consistency between content and style. Alternatively, we used discriminator $D$ for adversarial training. The basic function of discriminator $D$ is to identify and classify input images as either real or fake.

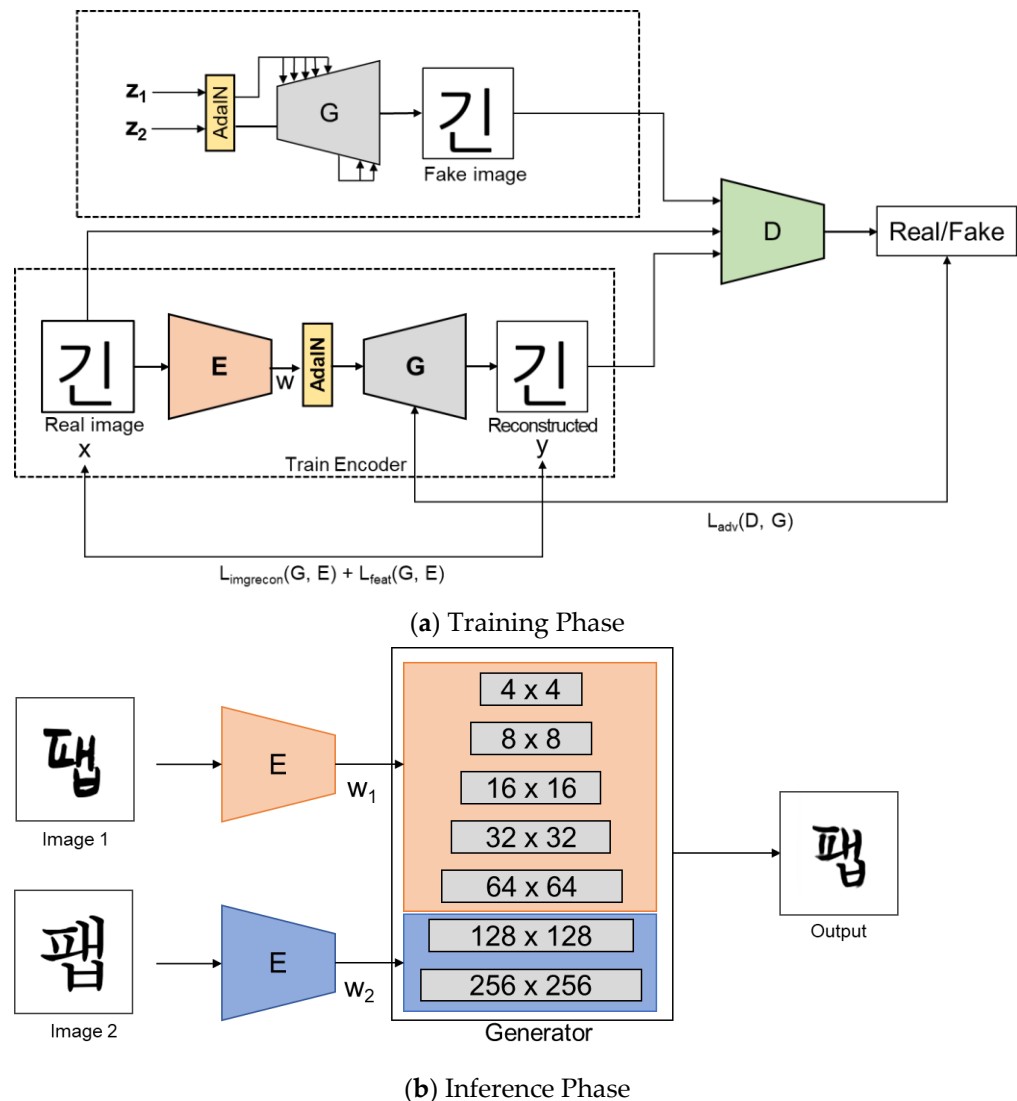

**Figure 2.** Main architecture of FFGAN; (**a**) training strategy for style mixing and image reconstruction; (**b**) Inference stage. Encoder *E* extracts style latent vectors from input images and sends them as input to the generator.

### 2.2. Font Fusion at Inference

Once trained, the FFGAN model enables the fusion of diverse font collections, which include a combination of handwritten and printed font styles, achieved through the implementation of mixing regularization techniques. To achieve this, we extracted style vectors $W^1$ and $W^2$ from a handwritten image (image 1) and a printed image (image 2), respectively, as shown in Figure 2b.

Subsequently, we injected the $W^1$ vector into the first five layers and $W^2$ into the last two layers of the generator. The injection of style vectors at several layers guarantees the preservation of the intended content image while integrating stylistic qualities such as strokes and thickness from the style image.

### 2.3. Design of Fusion Encoder

The architectures of the proposed generator and discriminator are primarily inspired by the StyleGAN [24] model, with a few adjustments examined in Section 3.2. Various options must be considered when designing the network architectures for fusion encoders. Users are given the flexibility to select specialized feature extractors such as VGG [26] or ResNet [27], or they can explore innovative encoder architectures designed specifically for

extracting style code vectors from images in image-to-image translation scenarios [9,10,25]. In this section, we investigate the impacts of different fusion encoder design choices.

We initially experimented with the VGG architecture using separate and joint training. However, this approach was susceptible to overfitting. Despite efforts to fine-tune hyperparameters, the model still struggled to mitigate overfitting. Subsequently, we replaced the fusion encoder in our system with a style encoder inspired by FUNIT [9]. This configuration is referred to as a Style Coder. The Style Coder was trained in combination with both the generator and discriminator.

Experiments were conducted, but the findings did not align with our expectations. Consequently, we concluded the impact of our selection of the fusion encoder design on the overall performance. The replacement of the proposed fusion encoder with a style encoder resulted in a decline in performance. The chosen fusion encoder plays a crucial role in improving the overall quality of the mixed-generated images.

### 2.4. Loss Functions

The loss functions used in our study were used to facilitate the development of controllable-style mixing for font regularization. These objective functions are further discussed in the following sections.

#### 2.4.1. Adversarial Loss

The non-saturating GAN loss technique was used in the adversarial training process involving the generator $G$ and discriminator $D$. The role of the discriminator is to classify the generated fake image ($y$) produced by the generator, which can be randomly generated from a Gaussian distribution ($z^1$, $z^2$) or reconstructed from the input image ($x$) as a fake image after passing through a transformation ($x$ -> $W$). In contrast, the generator aims to deceive the discriminator by synthesizing images that appear realistic.

$$min_D max_G L_{adv}(D, G) = E_x[\log D(x)] + E_y[-\log D(y)]. \tag{1}$$

#### 2.4.2. Image Reconstruction Loss

The loss function facilitates the image reconstruction learning process. The *L1* distance measurement was used between the reconstructed image $y$ and the corresponding real image $x$ to reduce the pixel-level difference. During the joint training process, generator $G$ and encoder $E$ were subjected to this loss. The loss function is expressed as follows:

$$L_{imgrecon}(G, E) = E_{x,y}[||x - y||_1]. \tag{2}$$

#### 2.4.3. Feature Matching Loss

In our research, we observed that using pixel-level loss alone often leads to the production of blurry images. To overcome this issue, we incorporated feature-matching loss into our approach. To this end, we utilized discriminator $D$ as a feature extractor, which we refer to as $D^f$, and compared the feature difference using $L_1$ distance. This comparison enabled us to minimize the variation between the two images. We applied this loss to both generator $G$ and encoder $E$. Thus, our enhanced loss formulation helps generate smoother and more visually appealing images.

$$L_{Feat}(G, E) = E_{x,y}\left[\left|\left|D^f(x) - D^f(y)\right|\right|_1\right]. \tag{3}$$

The main learning objective of the proposed FFGAN can be stated as follows:

$$min_D max_G \lambda_{adv} L_{adv}(D, G) + \lambda_{imgrecon} L_{imgrecon}(G, E) + \lambda_{feat} L_{feat}(G, E) \tag{4}$$

where $\lambda_{adv}$, $\lambda_{imrecon}$ and $\lambda_{feat}$ are the tunable hyperparameters.

## 3. Experiment

In this section, we validate the proposed FFGAN model by focusing on its performance in a mixing task involving Korean and Chinese characters.

### 3.1. Datasets

We created a dataset consisting of 174 Google fonts for the Korean Hangul font-mixing task to ensure an equal representation of handwritten and printed fonts. Each font included the 2350 most common Korean characters, with 80% of the dataset dedicated to training, encompassing approximately 2000 characters, and the remaining 20% for testing. Additionally, during testing, 20% of the 179 fonts were used for inference validation to assess task performance across diverse font styles.

To evaluate the Chinese font-mixing task, we assembled a diverse dataset sourced from FreechineseFont, encompassing 110 unique font styles equally split between 55 handwritten and 55 printed styles. Every font in the dataset included the most frequently used 1000 Chinese characters. Our data distribution allocated 80% of the fonts for training, where each font had 800 characters, and the remaining 20% for testing. Within the test set, 200 characters from the fonts were not observed during training, ensuring a comprehensive assessment of the generalization ability, and 20% of the test fonts were reserved for this purpose.

### 3.2. Training Details

The StyleGAN-V2 [24] model forms the foundation for generator G and discriminator D, with several modifications made to the specific aspects mentioned. The nonlinear mapping network was eliminated from the network. The dimensions of latent vectors $z^1$ and $z^2$ were set to 64. The dimensions of the generator $G$ and discriminator $D$ channels were halved. Regularization was not employed in the implementation of $G$. Discriminator $D$ also performs feature extraction from the intermediate layers to train fusion encoder $E$ and generator $G$. The weight assigned to each loss function, namely $\lambda_{adv}$, $\lambda_{imrecon}$, and $\lambda_{feat}$ in Equation (4), was uniformly set to one, denoted as $\lambda = 1$. In addition, we maintained the default configuration of StyleGAN-V2, including the utilization of R1 regularization [28], the Adam optimizer [29], predetermined learning rates, and the exponential moving average technique for the generator. In the process of fusion encoder E, we used an architecture identical to that of discriminator $D$ with up to $8 \times 8$ convolutional layers. In addition, we introduced an average pooling layer that demonstrates stability and generalization. Minibatch discrimination [30] was not employed by the fusion encoder $E$, in contrast to discriminator $D$.

To train the entire model, we used a combined training approach involving a fusion encoder $E$, generator $G$, and discriminator $D$. This method was developed to address the challenge of not providing clear guidance at both image and set levels (unsupervised). To help the generator effectively mix different styles, we introduced two random vectors, $z^1$ and $z^2$, chosen randomly from a Gaussian distribution. We applied $z^1$ before a certain point (the initial five layers of the generator) and $z^2$ after it (the last two layers of the generator), as shown in the upper box of Figure 2a. Injecting $z^1$ in the lower-resolution parts of the model influenced the overall shape of the glyph, whereas injecting $z^2$ in the higher-resolution sections affected style elements such as thickness, strokes, and serifs. Moreover, when the style vector code $W$ was obtained from the fusion encoder $E$ during the learning process to map the input image into the corresponding latent space for accurate reconstruction, it was directly injected into all layers of G, as shown by the lower box in Figure 2a. By training a GAN on a style mixing technique, it learns a diverse style of handwritten and printed font simultaneously, and their disentanglement. This technique was first introduced in StyleGAN to generate high-quality human facial images. During training, encoder $E$ learns to project an input image into a latent space to obtain its latent vector. We trained encoder $E$ to reconstruct the same input image by passing the latent vector of the input image onto the generator. At the end of the training, our encoder learns how to project the input images into the latent space, and our generator can either generate an image from two noise

vectors, $z^1$ and $z^2$, randomly chosen from a Gaussian distribution, or reconstruct the image from the output of the encoder, which is a latent vector of an image. During the inference stage, we can control the input images; therefore, using our trained encoder, we pass the two input images, one of handwritten font and the other of printed font, to obtain latent vectors, and both latent vectors pass through the generator. The handwritten font image latent vector passes through the initial five layers of the generator, and the printed font image latent vector passes through the last two layers of the generator to obtain a refined handwritten font image.

*3.3. Qualitative Results*

The visual outcomes are illustrated in Figures 3 and 4, which show the qualitative results. We generated diverse Korean and Chinese characters by employing various font styles. The key concept of the FFGAN model is to achieve disentanglement (style mixing) between handwritten and printed fonts using mixing regularization in a completely unsupervised manner. In the inference stage, fusion encoder *E* plays a crucial role by extracting the style code vector from both handwritten and printed input font images. This enables the generator to perform style mixing. We observed that the proposed approach successfully separated style and content, showcasing its effectiveness in generating mixed or refined handwriting images.

**Figure 3.** Illustrates a font fusion example. FFGAN demonstrates its ability to effectively produce a mixed image (output) that combines the stylistic elements of both a handwritten and printed image.

**Figure 4.** Represents the output of Chinese characters.

*3.4. Baseline Evaluation*

In this study, we conducted a comparative analysis of two advanced font generation (FG) methods. Certain methods were excluded from the comparison for specific reasons. First, we excluded methods that are not suitable for handling font styles that have not been seen before and would require extra adjustments, such as fine-tuning [2,3,7]. Second, we removed the non-generic models created for specific compositional writing systems [1]. Another clarification is that our model is specifically designed for font-mixing in the one-to-one mapping of printed characters and handwritten characters, and our output is a mixed generated character, whereas other methods are either performing FG or style transfer. Therefore, for the comparison analysis, we attempt to transfer the style through FFGAN using style mixing, and we also obtain good results compared to baselines. Our model is similar to the FMGAN [31] model because it is a modified version of StyleGAN, with modifications owing to font image generation. The main difference between FMGAN and FFGAN is the mapping network, the loss functions, and FMGAN are primarily used for font image generation or style transfer. Figure 5 presents the results of the visual analysis.

**Figure 5.** Visual comparison against the baselines.

The comparative assessment was focused on our model against two FG methods: FUNIT [9] and FMGAN [31]. We used the official implementations of FUNIT and FMGAN, which were trained from scratch. For a fair comparison, we used the same starting font when testing all methods. We also evaluated them under similar conditions using numerous examples to ensure fairness. We refrain from directly comparing our model to other state-of-the-art font generation models. This decision stems from the fact that our primary objective diverges from theirs. While they concentrate on generating fonts using various approaches, our focus revolves around the enhancement of handwritten fonts through the implementation of a unique mixing regularization technique.

Our approach is distinct and centers on the refinement of handwritten fonts. Rather than competing with existing font generation models, we aim to augment the quality and appeal of handwritten text styles. This sets us apart from the conventional path of font generation, allowing us to explore innovative methods and techniques tailored specifically for the improvement of handwritten fonts. By adopting a mixing regularization technique, we are able to introduce a novel dimension to the art of font refinement. This technique empowers us to fine-tune and perfect the subtleties of handwritten characters, resulting in fonts that possess a distinct, polished charm. While other models may excel in font creation, we have chosen to carve a unique niche in the realm of handwritten font enhancement, pushing the boundaries of what can be achieved in this specialized field.

*3.5. Quantivate Results*

We conducted experiments to obtain quantitative outcomes from an average of six separate experiments in which eight types of font styles—combinations of Korean and Chinese, handwritten and printed, and Gothic and Ming—were used to compare the results. Table 1 lists the learned perceptual image patch similarity (LPIPS) of the generated font images. LPIPS is a perceptual similarity metric used to access the visual quality of images generated by a GAN by measuring their similarity to reference images based on deep feature extraction and perceptual distance computation. We compared the scores of our model with those of FUNIT and FMGAN, as these types of metrics require ground truth values. In this study, we used printed images as the ground truth, and the state-of-the-art methods attempted to transfer the style of printed images to handwritten images. In some cases, the result is better than FFGAN because our proposed model fuses the two fonts for refinement, and it is not exactly similar to the printed font image. Table 2 lists the Fréchet inception distance (FID) score, which is a metric used to assess the quality of images generated by measuring the similarity between the distribution of features extracted from the generated and reference images using a pretrained Inception V3 neural network [21]. We compared our model with FMGAN because FFGAN and FMGAN generate images using style mixing based on the StyleGAN model, whereas FUNIT is a pure image-to-image translation model used to generate a content image with the style of the reference image. Therefore, FUNIT requires ground truth images to measure the FID score, and ground truth was not available in this study. Similarly, to calculate evaluation metrics for other state-of-the-art FG models, we also need ground truth. However, in our case, as the generated output is a fused image, no ground truth is available. Therefore, it is not a fair comparison of our model with other state-of-the-art FG models because other models are used for the FG task, whereas the main motivation of our model is to refine the handwritten fonts.

**Table 1.** Quantitative comparison of LPIPS scores between the proposed method and the baselines.

|  | Korean (LPIPS) | | Chinese (LPIPS) | |
| --- | --- | --- | --- | --- |
|  | Seen Characters | Unseen Characters | Seen Characters | Unseen Characters |
| FUNIT | 0.14 | 0.18 | 0.21 | 0.26 |
| FMGAN | 0.9 | 0.12 | 0.12 | 0.14 |
| FFGAN | 0.8 | 0.11 | 0.9 | 0.13 |

**Table 2.** Quantitative comparison of FID score of the proposed method and baseline.

|  | Korean (FID) | | Chinese (FID) | |
| --- | --- | --- | --- | --- |
|  | Seen Characters | Unseen Characters | Seen Characters | Unseen Characters |
| FMGAN | 21.44 | 21.75 | 61.85 | 72.55 |
| FFGAN | 20.15 | 20.98 | 22.86 | 23.14 |

## 4. Conclusions

We present FontFusionGAN (FFGAN), a novel and highly effective approach for enhancing handwritten fonts through incorporating style mixing techniques. We address the inherent challenges associated with handwritten fonts, particularly their legibility and consistency. Although many existing methods focus on printed fonts, our research places significant emphasis on refining handwritten fonts, rendering it a pioneering effort in this domain. By adopting mixing regularization, style localization was enabled in the font fusion process. This innovation involves generating images during training using two random latent codes instead of one, leading to the synthesis of fonts that seamlessly blend the desirable features of both handwritten and printed styles. Our evaluations and experiments using FFGAN on a dataset of handwritten fonts demonstrate remarkable

improvements in font legibility and aesthetic appeal. The ability of the model to generate fused font images that surpass the quality of the original handwritten fonts while preserving their unique styles holds significant promise. Our approach functions in an unsupervised manner, eliminating the need for content-based or class-label-based supervision, making it adaptable and efficient for a wide range of applications. We tested the proposed model using our handwritten images, which showed appealing results for the refined images, as shown in Figure 6. This also proves the few-shot learning behavior of the proposed model. One of the main limitations of our model arises during the inference stage because it requires pairs of images with the same content to generate high-quality fused images. In addition, our model cannot be effectively used to generate fonts independently because the quality of the generated fonts remains suboptimal unless they are fused with pre-existing printed fonts during testing.

**Figure 6.** Real handwriting of a human (from our team). We mix the handwriting using FFGAN with printed fonts and obtain the refined font.

**5. Future Work**

We are currently in the process of preparing a Handwriting Font Fusion Service, which will become available in the near future. Our plan involves optimizing the parameters of both the generator and discriminator. Currently, we utilize the generator and discriminator from StyleGAN, which is a resource-intensive model that requires significant training time. Furthermore, we aim to enhance the ability of our fusion encoder to decode both the content and style from images, thereby reducing its dependence on images with identical content.

**Author Contributions:** Conceptualization, A.K. and K.K.; methodology, A.K., K.K. and A.u.H.M.; software, A.K. and K.K.; validation, A.u.H.M. and J.C.; formal analysis, A.K.; investigation, A.K.; resources, J.C.; data curation, A.K.; writing—original draft preparation, A.K.; writing—review and editing, J.C., A.u.H.M. and A.K.; visualization, A.K.; supervision, J.C.; project administration, J.C.; funding acquisition, J.C. All authors have read and agreed to the published version of the manuscript.

**Funding:** This research received no external funding.

**Data Availability Statement:** Some or all data, trained models, or codes that support the finding of this study are available from the corresponding author upon request.

**Conflicts of Interest:** The authors declare no conflict of interest.

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
