# Peer review of "FontFusionGAN: Refinement of Handwritten Fonts by Font Fusion"

_electronics, doi:10.3390/electronics12204246_

Round 1

Reviewer 1 Report

The Introduction is lacking more recent and pertinent research findings/references from the topic. The authors are advised to revise the Introduction section by adding relevant citations and by making a comparison with the present study in terms of efficacy and benefits.

Please highlight the major shortcomings and limitations of the proposed approach.

The fonts in the text and tables is not uniform throughout the manuscript. Please correct it.

The quality of Figure 1 must be improved. It is not readable and clear.

The English language must be revised, as well as the figure captions. There are multiple grammatical errors and typos throughout the manuscript.

Please revise the Conclusions by highlighting the important outcomes, their significance, and take-home-message from the paper.

The English language must be revised, as well as the figure captions. There are multiple grammatical errors and typos throughout the manuscript.

Reviewer 2 Report

The paper contributes to solving an interesting problem of handwritten text refinement, which has several useful applications. The following suggestions can help enhance the manuscript further.

The abstract does not adequately summarize the entire manuscript as the abstract details of methodology and results are not provided.

Too many details are given in the caption of Figure 1. Figure captions must be brief. The figure must be explained in the main text, not the caption.

Referring to Fig 1., it is not clear how the two stages of the proposed methodology given in Fig. 1 (a) and (b) work together. A clear connection between them is missing in the description.

The section “Design of Fusion Encoder” under the experimentation section is a part of the methodology. Hence, in my opinion, it may moved under the methodology section.  

Did the study find any issues with the fused fonts produced by the proposed technique? If yes, please describe them in the results section.

Give any limitations of the study in the conclusion section.

All abbreviations must be introduced in their first appearance with their full form. The evaluation metric FID has not been introduced before use.

The language of the manuscript is generally acceptable. Some minor editing may be required.

Reviewer 3 Report

The paper introduces a FFGAN based handwritten fonts generation model. The idea is interesting and the organization is appropriate. However, there is some confusion of the proposed model. Some comments are not convincing. This makes unclear to the readers the need and benefits of the proposed methodology.

1. The motivation is not clear enough. In my opinion, existing handwriting fonts generation methods are able to meet most demands. Why this fusion is necessary to generate new fonts? Besides, will it cause lower-quality generation results?

2. The authors only provide a single comparison related to GAN as FMGAN. It is not the SOTA generation model. More GAN-based advanced generation models and other models such as VAE-based method and diffusion models should be compared. Moreover, according to the results, the proposed FFGAN method did not show obvious improvement compared with FMGAN. More illustrations should be provided.

3. The description of the experimental subject is lacking. more details should be discussed.

 Moderate editing of English language required

Round 2

Reviewer 2 Report

The authors have addressed all my concerns from the first review and the paper can be accepted for publication in the current form. 

Reviewer 3 Report

The authors have addressed my concerns